# Single-Cell RNAseq Profiling of Human γδ T Lymphocytes in Virus-Related Cancers and COVID-19 Disease

**DOI:** 10.3390/v13112212

**Published:** 2021-11-03

**Authors:** Juan Pablo Cerapio, Marion Perrier, Fréderic Pont, Marie Tosolini, Camille Laurent, Stéphane Bertani, Jean-Jacques Fournie

**Affiliations:** 1Centre de Recherches en Cancérologie de Toulouse, Inserm UMR 1037, 31037 Toulouse, France; marion.perrier@inserm.fr (M.P.); frederic.pont@inserm.fr (F.P.); marie.tosolini@inserm.fr (M.T.); camille.laurent@inserm.fr (C.L.); 2Toulouse University, 31062 Toulouse, France; stephane.bertani@ird.fr; 3CNRS UMR 5071, 31024 Toulouse, France; 4Institut Universitaire du Cancer-Oncopole de Toulouse, 31100 Toulouse, France; 5Laboratoire d’Excellence ‘TOUCAN-2’, 31037 Toulouse, France; 6Institut Carnot Lymphome CALYM, 75014 Paris, France; 7Centre Hospitalier Universitaire, 31300 Toulouse, France; 8PHARMADEV, IRD UMR 152, 31062 Toulouse, France

**Keywords:** human, gammadelta, lymphocyte, tumor, COVID-19, transcriptome, single cell, differentiation, trajectory

## Abstract

The detailed characterization of human γδ T lymphocyte differentiation at the single-cell transcriptomic (scRNAseq) level in tumors and patients with coronavirus disease 2019 (COVID-19) requires both a reference differentiation trajectory of γδ T cells and a robust mapping method for additional γδ T lymphocytes. Here, we incepted such a method to characterize thousands of γδ T lymphocytes from (*n* = 95) patients with cancer or adult and pediatric COVID-19 disease. We found that cancer patients with human papillomavirus-positive head and neck squamous cell carcinoma and Epstein–Barr virus-positive Hodgkin’s lymphoma have γδ tumor-infiltrating T lymphocytes that are more prone to recirculate from the tumor and avoid exhaustion. In COVID-19, both TCRVγ9 and TCRVγnon9 subsets of γδ T lymphocytes relocalize from peripheral blood mononuclear cells (PBMC) to the infected lung tissue, where their advanced differentiation, tissue residency, and exhaustion reflect T cell activation. Although severe COVID-19 disease increases both recruitment and exhaustion of γδ T lymphocytes in infected lung lesions but not blood, the anti-IL6R therapy with Tocilizumab promotes γδ T lymphocyte differentiation in patients with COVID-19. PBMC from pediatric patients with acute COVID-19 disease display similar γδ T cell lymphopenia to that seen in adult patients. However, blood γδ T cells from children with the COVID-19-related multisystem inflammatory syndrome are not lymphodepleted, but they are differentiated as in healthy PBMC. These findings suggest that some virus-induced memory γδ T lymphocytes durably persist in the blood of adults and could subsequently infiltrate and recirculate in tumors.

## 1. Introduction

Efficient protection against viruses and cancers requires innate and adaptive immunity. Due to their ontogeny, antigen recognition mode, and effector functions, γδ T lymphocytes are involved in both types of immune defense [1]. As a result, γδ T cells are activated by a wide array of viral infections as well as by many different types of cancer. Human CD4^−^CD8^−^ γδ T lymphocytes express either TCRVδ2 or TCRVδ1/3 chains prominently paired with TCRVγ9 or TCRVγ2/3/4/5/8 (TCRVγnon9), respectively. The TCRVγ9Vδ2 γδ T cells are prominently circulating lymphocytes specifically reactive to phosphoantigen-sensing butyrophilins while TCRVγnon9 γδ T cells are mostly tissular and recognize more diversified antigens. Upon antigen activation, both subtypes differentiate from naive (Tn) to central memory (Tcm), effector memory (Tem), and terminally differentiated effector memory CD45Ra+ (Temra) cells [2]. Regardless of the γδ TCR subset, the strong cytotoxic properties of γδ Tem cells play a crucial role in the killing of virus-infected and cancer cells. For decades, TCRVγ9Vδ2 γδ T cells have been known to react to cells infected by viruses such as the Epstein–Barr virus (EBV) [3], herpes simplex viruses [4], human immunodeficiency virus (HIV) [5], [6], human papillomavirus (HPV) [7], hepatitis B virus (HBV) [8], Influenza viruses [9,10], dengue virus [11], and severe acute respiratory syndrome-related coronavirus (SARS-CoV) [12]; whereas the TCRVγnon9 lymphocytes [13] respond, inter alia, to cytomegalovirus (CMV), hepatitis E virus (HEV) [14], and Kaposi’s sarcoma-associated herpesvirus (HHV-8) [15].

The overall T and γδ T cell response remain hitherto poorly understood in coronavirus disease 2019 (COVID-19) caused by severe acute respiratory syndrome coronavirus 2 (SARS-CoV-2) infection. Like other respiratory viral infections, COVID-19 can exhibit distinct patterns of clinical features and disease severity, ranging from a virus-induced respiratory pathology due to a weak immune response to an overaggressive immunopathology caused by excessive immune activation. The heterogeneous immune profiles recently characterized in peripheral blood mononuclear cells (PBMC) of COVID-19 patients reflect this wide range of symptoms. COVID-19 patients have a quantitatively decreased compartment of γδ T cells that are more differentiated than control samples [16], accompanied by an increased number of immature neutrophils [17]. Similar alterations were reported with the other adaptive and unconventional T lymphocytes [18,19], supporting a scenario of broad T cell immunosuppression linked to fatal outcomes [20].

Likewise, the extent and quality of tumor-infiltrating T lymphocytes (TILs) in cancer are important determinants of the outcome of immunotherapy or untreated cancer patients. On the one hand, a weak immune response to cold tumors, peripherally excluding tumors, or tumors infiltrated by exhausted T lymphocytes allows for uncontrolled cancer and leads to fatal outcomes. On the other hand, too strong immune reactions are sometimes observed in untreated cancers (for review [21]) and systematically in patients receiving immune checkpoint blockade. Therefore, viral infections and cancer can present large and heterogeneous spectrums of immune response patterns. In human cancer, our research and other studies have shown that the abundance of γδ TILs, whether TCRVγ9 or TCRVγnon9, varies substantially across individuals and among cancer types, and that it is associated with patient outcomes [22,23]. As opposed to peripheral blood and hematopoietic malignancies dominated by TCRVγ9 γδ T cells, the predominant subset of γδ TILs in most solid tumors are TCRVγnon9 cells, whose number, cytotoxic differentiation, and functional status vary between individuals [24,25,26].

We recently depicted the pan-cancer landscape of human γδ TILs using single-cell RNA sequencing (scRNAseq). This strategy unveiled that the differentiation of cytotoxic γδ T cells from the TCRVγ9 and TCRVγnon9 subsets was related to the CMV status of the healthy donors [27]. Furthermore, despite the considerable heterogeneity of cancer patients, their infiltration and differentiation appeared phased with that of the adaptive T CD8 TIL compartment, yielding correlated rates of functional and/or exhausted γδ T and T CD8 TILs in human cancers [28].

Overall, these findings raised questions as to whether the viral status of cancer patients affected their γδ TILs, and whether the transcriptomic profiles correlated with those of T lymphocytes from COVID-19 patients’ PBMC. Here, we addressed these questions through a comprehensive characterization of the differentiation and functional hallmarks of γδ T lymphocytes by analyzing scRNAseq datasets of tumors from cancer patients with known viral status, and PBMC from COVID-19 patients.

## 2. Materials and Methods

### 2.1. scRNAseq Datasets Pre-Processing

The (cells, gene) matrix data for the control γδ T isolated from healthy donor’s PBMC has been previously published [28]. The scRNAseq of tumor lesions from Hodgkin’s lymphoma (HL) (EGAS00001004085) [29], head and neck squamous cell carcinoma (HNSCC) (GSE139324) [30], PBMC from COVID-19 patients (GSE145926, GSE155224, GSE155249, GSE166489, and GSE167029) were downloaded and assembled to digitally extract their γδ T lymphocytes. After pre-processing and discarding cell doublets and dying cells, all datasets were integrated using the R package Seurat version 4.0 [31]. Principal components analysis (PCA) was performed on this integrated dataset before uniform manifold approximation and projection (UMAP) [32] to check the integrated dataset quality.

### 2.2. Single-Cell Signatures and Scores

After normalization and integration, the integrated scRNAseq dataset was scored for a collection of multigene signatures [28] using Single-Cell Signature Explorer software [33]. Briefly, the score of each single-cell C_j_ for gene set GS_x_ was computed as the sum of all unique molecular identifiers (UMI) for all the GS_x_ genes expressed by C_j_ divided by the sum of all UMI expressed by Cj:(1)$$\mathrm{Score}\;\mathrm{of}\;\mathrm{cell}\;\mathrm{Cj}\;\mathrm{for}\;\mathrm{geneset}\;\mathrm{GSx}\;=\left(\sum \left(\mathrm{UMI}\right)\mathrm{GSx}\right)\mathrm{Cj}\;/\;\left(\sum \left(\mathrm{UMI}\right)\mathrm{Cj}\right)$$

All cell signature scores were merged with each cell’s trajectory coordinates using Single-Cell Signature Merger software [33]. For visualization, Single-Cell Multilayer Viewer was used, a serverless software allowing to merge up to five layers of quantitative and qualitative variables, alone or in combination [28].

### 2.3. Score and Gate for Digital Extraction of γδ T Lymphocytes

Starting from any scRNAseq datasets pre-processed as above, this procedure comprised five gating steps sequentially applied using Single-Cell Virtual Cytometer software [34]:Positive selection of double negative cells (= T and NK lineages) from the scatterplot of ‘B cell’ and ‘Myeloid cell’ signature scores.Scatterplot of ‘B cell’ signature against the addition of ‘CD8AB’ and *‘TCRγδ’* signatures to positively select the double-positive cells (= non-CD4 T cells).Scatterplot of ‘CD8ab’ signature against ‘TCRα constant gene *TRAC*’ signature to digitally extract double-positive cells (= αβ T CD8 lymphocytes) on the one hand, and negative cells (= γδ T plus CD4CD8-double negative αβ T cells) on the other hand.Scatterplot of the addition of ‘CD3 complex’ and ‘TCRα constant gene *TRAC*’ signatures against the addition of ‘CD3 complex’ and ‘TCRγδ’ signatures to extract the ‘CD3 complex’ and ’TCRγδ’ double-positive cells (=γδ T lymphocytes).

The datasets of the finally identified γδ T cells were digitally extracted from the original scRNAseq dataset and further subtyped and injected on the trajectory (see below).

### 2.4. Subtyping of the TCRVγ9 and TCRVγnon9 γδ T Cells

This was performed as previously described [28]. Briefly, γδ T lymphocytes express either the *TRGC1*-encoded TCRVγ9 or the *TRGC2*-encoded TCRVγnon9 in a mutually exclusive fashion, so the extracted γδ T lymphocytes were categorized as TCRVγ9 cells based on (‘*TRDC, TRGC1*’ positive cells) or TCRVγnon9 cells based on (‘*TRDC, TRGC2’* positive cells) classifications using a compensated score of these two signatures. Compensated scores were obtained by multiplying the score of the gene set of interest, here GSS_TCRVγ9_, by its difference to its complementary gene set (GSS_TCRVγ9_ minus GSS_TCRVγnon9_):

GSS_TCRVγ9_*comp =* GSS_TCRVγ9_ × (GSS_TCRVγ9_ − GSS_TCRVγnon9_)(2)

Scatterplots of GSS_TCRVγ9_ and GSS_TCRVγnon9_ finally identified the TCRVγ9 or TCRVγnon9 subset of each γδ T lymphocyte [28].

### 2.5. Cells Injection on Public Pseudotimed Differentiation Trajectory

The maturation trajectory of γδ T cells was computed by minimum spanning tree and visualized as pseudotimed trajectories. Pseudotimed trajectories are differentiation trajectories shown with cell pseudotime on the *x-*axis and a projection of both trajectory dimensions (here MST1 and MST2) on the *y-*axis [28]. We will refer below to them as ‘public γδ T lymphocytes trajectory’ the pseudotimed differentiation trajectory of all γδ T lymphocytes from ~150 tissues samples, including healthy donors’ PBMC and cancer patients’ tumors (the (cell, gene) matrix data is available from [28]). New γδ T lymphocytes digitally extracted from additional datasets, such as those from COVID-19 patients, were injected onto this ‘public γδ T cell trajectory’. Briefly, a total of (*n* = 9) datasets for COVID-19-derived tissues were downloaded. These COVID-19 datasets were first filtered for quality control, integrated as described above, and their γδ T lymphocytes were digitally extracted by score and gate [28]. These new γδ T lymphocytes were then injected onto the public trajectory by using Seurat’s Integration and Map-to-Query workflow to determine their respective (MST1, MST2) coordinates and then infer their pseudotime within the public trajectory. This procedure encompasses the following four steps:

Step 1:Query 1 = γδ T lymphocytes digitally extracted from COVID-19 samples;Reference = public trajectory;Integration of gene expression data from Query 1 and Reference (Seurat’s Integration).

Step 2:Query 2 = Integrated (Query and Reference);Reference = public trajectory;Transferring MST coordinates from Reference metadata to Query 2 (Seurat’s Map-to-Query).

Step 3:Query 2 = pseudotime calculation using MST coordinates and R package Dynverse.

Step 4:Visualization of new data (Query 1) on the public trajectory (Reference) using Single-Cell Multilayer Viewer [28].

### 2.6. Classifications

The TILs classifications as Ttrm/non-Ttrm and Tex/non-Tex were performed by ‘at least one binary’. Briefly, the single cells were scored for several reference gene signatures as previously described [28]. For Tex classification, the reference Tex signatures were based on five published and partially overlapping Tex gene sets [35,36,37,38,39]. For each signature, a cut-off was defined as the maximal score of the (*n* = 3680) control γδ T lymphocytes extracted PBMC of healthy adults. This threshold defined the cell’s binary (1: cell score > threshold, 0: otherwise). The Tex cut-offs were: 3.9 for ‘Chihara_IL27_Coinhib_module’ [35]; 0.52 for ‘Alfei_d20_tox’ [36]; 0.22 for ‘Khan_Tox_OverExpressed_genes’ [37]; 0.5 for ‘Tosolini_ NHL_IEGS33’ [38]; and 0.16 for ‘Balanca_QP_genes’ [39]. For each TIL, the five Tex binaries were summed, and if the sum was non-zero, the TIL was classified as Tex, or otherwise as non-Tex. The Ttrm classification was applied likewise using six references and partially overlapping Ttrm signatures published previously [40,41]. The binarizing cutoffs established with the control γδ T lymphocytes were: 0.46 for ‘Kumar_13g_Ttrm’; 0.3 for ‘Kumar_3g_Ttrm’ [40]; 3.72 for ‘Wu_Tcellcluster4.1_trm’; 2.6 for ‘Wu_Tcellcluster8.3_trm’; 2.8 for ‘Wu_Tcellcluster8.3b_trm’; 4.2 for ‘Wu_Tcellcluster8.3c_trm’ [41]. These Ttrm binaries were summed, and if the sum was non-zero, the TIL was classified as Ttrm, or otherwise as non-Ttrm (recirculating).

## 3. Results

### 3.1. Tumor Infiltrating γδ T Lymphocytes from Virus-Positive and -Negative Cancer Patients

We downloaded the published scRNAseq dataset from a total of (*n* = 26) human tumors of head and neck carcinoma (HNSCC), including (*n* = 8) samples positive and (*n* = 18) negative for HPV, as well as (*n* = 9) Hodgkin’s lymphoma (HL) of mixed cellularity subtype in which (*n* = 5) samples were positive and (n = 4) negative for EBV. Their respective γδ T lymphocytes were digitally purified, and their TCR subtype and differentiation stage were determined by injection onto the ‘public γδ T cell trajectory’ (see Methods) and cross-labeling with reference gene signatures from external single-cell datasets of human TILs [28].

A total of (*n* = 1055) γδTILs ((*n* = 401) TCRVγ9 plus (*n* = 654) TCRVγnon9 cells) were extracted from HNSCC, and (*n* = 113) γδ TILs ((*n* = 51) TCRVγ9 plus (*n* = 62) TCRVγnon9 cells) from HL. In contrast with PBMC from healthy donors, the cell counts extracted from tumors varied considerably across individuals and in both diseases (range: 1–33 γδ TILs in HL patients, and 0–111 in HNSCC patients), as previously reported in other cancer types [28]. On average, HNSCC comprised more γδ TILs (mean = 40 cells, 15 TCRVγ9 + 25 TCRVγnon9) than HL (mean = 13 cells, 6 TCRVγ9 + 7 TCRVγnon9; *t*-test *P* = 3 × 10^−4^). The rates of their Tn, Tcm, Tem, and Temra stages varied between patients but without reaching significant difference between cancers and viral status (mean in HNSCC: 15% Tn, 37% Tcm, 47% Tem, 0.9% Temra; mean in HL: 9% Tn, 35% Tcm, 55% Tem, 0.8% Temra) (Figure 1a). In both cancers, these recirculating (non-Ttrm TILs, Methods) γδ TILs comprised less exhausted cells than their tissue-resident (Ttrm) counterparts (15% and 37%, respectively). Furthermore, the rates of recirculating γδ TILs were significantly associated with patient’s viral status (χ^2^ *P* value = 7.8 × 10^−18^): tumors of viral-positive HL and HNSCC included far more recirculating γδ TILs than viral-negative tumors (80% vs. 45% of γδ TILs, respectively) (Figure 1b).

Altogether, these data demonstrate that patients with HPV-positive HNSCC and EBV-positive HL have γδ TILs more prone to recirculate from the tumor and avoid exhaustion.

### 3.2. γδ T Lymphocytes from COVID-19 Patients

The scRNAseq datasets from PBMC of (*n* = 43) COVID-19 patients and (*n* = 21) healthy controls, as well as from broncho-alveolar lavage fluids (BALF) from (*n* = 20) COVID-19 patients and (*n* = 4) healthy controls were downloaded. A total of (*n* = 7473) γδ T lymphocytes were digitally extracted from these datasets, and their TCR subtype and differentiation stage were characterized as described above (see Methods).

In spite of similar dataset sizes, each sample of COVID-19 PBMC yielded on average three times less γδ T lymphocytes than the healthy controls (100 vs. 330, respectively, *t*-test P = 4 × 10^−5^), and both TCR subsets had decreased cell counts (on average 63 vs. 142 TCRVγ9 and 37 vs. 187 TCRVγnon9 in COVID-19 vs. control samples, respectively). Parallel analyses showed on average *n* = 561 and *n* = 479 T CD8 cells per sample in the same groups indicating that the COVID-19-related lympho-reduction from PBMC is specific to the γδ T cell lineage (Chi-2 *P* value = 5 × 10^−27^).

However, their differentiation stages remained unchanged between healthy and COVID-19 PBMC (7% Tn, 25% Tcm, 66% Tem, 3% Temra among TCRVγ9 cells; 19% Tn, 30% Tcm, 50% Tem, 2% Temra among TCRVγnon9 cell) (Figure 2a,b). In contrast, BALF of COVID-19 patients encompassed more γδ T lymphocytes than control BALF (on average 16 *vs.* 0 γδ T per BALF, respectively). Such COVID-19 BALF-derived γδ T encompassed both subsets (on average nine TCRVγ9 and seven TCRVγnon9 cells) and were more differentiated than in PBMC (on average 1% Tn, 18% Tcm, 81% Tem, 0% Temra among TCRVγ9 cells; 3% Tn, 36% Tcm, 61% Tem, 0% Temra among TCRVγnon9). Furthermore, the Ttrm gene signature was expressed by scarce (0.3%) γδ T lymphocytes in the control PBMC, but by 4% in COVID-19 PBMC and 17% in the COVID-19 BALF. Likewise, the Tex signature was up-regulated by 0%, 1%, and 24% γδ T lymphocytes, respectively, from these COVID-19 BALF samples (Figure 2a,b).

These data suggest that in adult COVID-19 patients, both subsets of γδ T lymphocytes relocalize from PBMC to the infected lung tissue, where their advanced differentiation, tissue residency, and exhaustion hallmarks witness of a recent T cell activation.

### 3.3. γδ T Lymphocytes in Clinical Groups of Adult Patients with COVID-19

These γδ T cell analyses were then extended to clinical subgroups of adult COVID-19 patients. The total number of γδ T lymphocytes in BALF of adult COVID-19 patients varied considerably between individuals without reaching a statistical difference between mild (*n* = 2) and severe (*n* = 6) disease (on average 39 vs. 22 γδ T cells per sample, respectively). The TCRVγ9 cell counts were similar in both groups (on average 13 vs. 14 cells per sample, respectively), whereas the TCRVγnon9 cell counts decreased in severe COVID-19 (on average 26 vs. 7) despite maintaining the same differentiation stage pattern (on average 1% Tn, 37% Tcm, 62% Tem, 0% Temra). Of note, with 0% Tn, 20% Tcm, 80% Tem, and 0% Temra in both groups, the TCRVγ9 cells were more differentiated than the TCRVγnon9 cells. The γδ T lymphocytes in BALF of patients with mild disease included 9% Ttrm and 12% Tex cells, contrasting to 40% Ttrm and 40% Tex in severe disease, and both subsets behaved similarly (Figure 3a). Hence, acute COVID-19 increases both tissue residency and exhaustion of γδ T lymphocytes in infected lung lesions.

The PBMC of adult COVID-19 patients treated (*n* = 12) or not (*n* = 4) by Tocilizumab, an anti-IL-6R mAb drug, had similar cell counts of total γδ T lymphocytes (on average 59 vs. 45 γδ T cells per sample, respectively), as for both TCRVγ9 and TCRVγnon9 subsets. Meanwhile, Tocilizumab clearly promoted both subsets’ differentiation (TCRVγ9 untreated: 8% Tn, 29% Tcm, 61% Tem, 2% Temra; TCRVγ9 treated: 2% Tn, 11% Tcm, 85% Tem, 2% Temra; TCRVγnon9 untreated: 30% Tn, 31% Tcm, 39% Tem, 0% Temra; both subsets treated: 9% Tn, 16% Tcm, 74% Tem, 1% Temra). This treatment did not affect their low rates of Tex cells (Figure 3b). Therefore, anti-IL6R treatment favors γδ T lymphocyte differentiation in COVID-19 patients.

### 3.4. γδ T Lymphocytes in Clinical Groups of Pediatric Patients with COVID-19

Finally, we analyzed the γδ T cell landscape in PBMC of children with acute COVID-19 (*n* = 5), multisystem inflammatory syndrome (MIS-C) (*n* = 14), and healthy pediatric controls (*n* = 7). The PBMC of pediatric controls comprised more γδ T lymphocytes of both TCR subsets than the PBMC of adult controls (*n* = 4) (average γδ T cell counts: 393 vs. 220, respectively), although their differentiation was similar (TCRVγ9: 5% Tn, 43% Tcm, 51% Tem, 2% Temra; TCRVγnon9: 14% Tn, 45% Tcm, 40% Tem, 1% Temra) (Figure 4a).

PBMC from acute and MIS-C COVID-19 children comprised less γδ T lymphocytes of both subsets than the pediatric controls (average γδ T cell counts: 71 (acute), 163 (MIS-C) *vs.* 393 (controls)). They were also less differentiated in acute disease than in MIS-C and pediatric controls (acute: TCRVγ9: 14% Tn, 52% Tcm, 34% Tem, 0% Temra; TCRVγnon9: 37% Tn, 44% Tcm, 19% Tem, 0% Temra; MIS-C: TCRVγ9: 10% Tn, 26% Tcm, 61% Tem, 3% Temra; TCRVγnon9: 22% Tn, 28% Tcm, 50% Tem, 1% Temra). Last but not least, the PBMC of pediatric controls contained no Tex cell among γδ T lymphocytes (in both subsets), as in the acute COVID-19 samples, but *n* = 2 Tex out of 2285 cells were detected in MIS-C samples (Figure 4b).

Together, these results indicated that acute COVID-19 disease induces a similar γδ T cell lymphopenia PBMC of pediatric and adult patients. In contrast, the PBMC-derived γδ T cells from MIS-C patients with SARS-CoV-2 infection are not lymphodepleted and resemble to those from healthy PBMC.

## 4. Discussion

In single-cell transcriptomics, pseudotime describes the progression of each cell alongside a continuous series of differentiation states, through a non-linear transformation of real chronological time [42,43,44]. How explicit time is quantitatively converted into γδ T cell pseudotime remains to be determined, and depends on the transcriptome diversity and the total number of single cells in the dataset. All the features of differentiation trajectories are strictly dependent upon the number of single cells selected. Hence, once a trajectory has been inferred from a given single-cell dataset, adding *ab extra* single cells to this dataset will change the trajectory, preventing their mapping on the former map. Here, we addressed this issue by incepting a method to inject without distortion any amount of *ab extra* cells on the reference trajectory. 

In this way, we were able to map and thus characterize thousands of γδ T lymphocytes from newer datasets onto a formerly-built reference map called here ‘public γδ T cell trajectory’ [28]. Of note, this trajectory encompasses a remarkable set of γδ T lymphocytes, being currently paved with ~30,000 γδ T cells from ~220 healthy and diseased adult individuals. As a result, it achieves a much higher level of exhaustivity and resolution than is possible with the cells from a single individual alone. Together, the public γδ T cell trajectory and the method for mapping newer γδ T cells provide a unique resource for characterizing these lymphocytes from diverse sources at the highest level of resolution.

As a result of high-resolution mapping of thousands of γδ T lymphocytes from patients with cancer or COVID-19, we were able to gain a wealth of information about γδ T lymphocyte differentiation dynamics at the single-cell scale. As reviewed in this series, the role of human γδ T cells in viral infections is highly investigated and debated in the context of SARS-CoV-2 infection [45,46,47,48]. Here, we found in PBMC of COVID-19 patients that γδ T lymphocytes were decreased compared to control samples, a finding consistent with flow cytometry studies [16,20,49,50]. Although a peripheral lymphopenia in COVID-19 was initially thought to indicate T cell immunosuppression, our present maps and other studies [18] show that γδ T cells increase concurrently in BALF of COVID-19 patients. T cell exhaustion is rarely observed among the γδ T cells in PBMC from adult and pediatric patients with COVID-19, even during its acute phase as reported with HEV infection [14]. Most Tex γδ T cells were instead found in the infected lung rather than in the circulating blood. Hence, COVID-19-induced γδ T cell exhaustion likely represents a scar of recent localized T cell activation in both TCRVγ9 and TCRVγnon9 subsets of cells rather than in a single subset alone [17]. Moreover, their advanced differentiation stages in blood from children with multisystem inflammatory syndrome associated with SARS-CoV-2 infection further support this view [51]. Tocilizumab, an IL-6 pathway inhibitor, effectively reduces mortality of severely ill and lymphopenic COVID-19 patients, notably by attenuating inflammation and normalizing their circulating T cell rates [52,53]. Consistent with these reports, the increased effector memory cells we observed here indicates that Tocilizumab also promotes γδ T cell differentiation, which constitutes a bioactivity of dual interest for both antiviral [45] and anticancer [54] immunity.

Single-cell studies have shown that the differentiation dynamics of human γδ T lymphocytes in both TCRVγ9 and TCRVγnon9 subsets primarily reflect the transcriptional emergence of their cytotoxic function, which peaks at the Tem and Temra stages [27]. In many solid and hematological human cancers, both lineages of γδ T and T CD8 TILs present strikingly coherent differentiation profiles, and a sizeable fraction of γδ T TILs are tissue-resident memory and exhausted cells [28]. Many T CD8 TILs in melanoma are bystander cytotoxic lymphocytes with a TCR specificity for viral antigens (V-spe), while only a minority represent true tumor-antigen-specific (T-spe) lymphocytes [55]. Among such CD8 TILs, the V-spe clonotypes prominently display functional Tcm and Tem differentiation stages whereas the majority of T-spe clonotypes are dysfunctional Tex cells [56]. In parallel, we found that HPV-positive HNSCC and EBV-positive HL patients have γδ TILs that are more prone to recirculate from the tumor and avoid exhaustion. Considering the above findings, it is tempting to hypothesize that a contingent of peripheral γδ T lymphocytes induced by viral infections does persist in the long term in the blood of adult individuals as central/effector memory cells, and can readily infiltrate tumors to mediate a large spectrum of HLA-unrestricted cytotoxic activities despite true antigen specificity.

For its clinical relevance to cancer immunotherapy, future studies from our team will seek to discern at the single-cell level the T-spe and V-spe clonotypes persisting among the human γδ T lymphocytes in the peripheral blood and infiltrating tumors. Further, our study provides a fundamental framework for characterizing γδ T lymphocytes in viral diseases and cancers, and provides the community with a unique resource for the monitoring of human γδ T lymphocytes at high-resolution.

## Figures and Tables

**Figure 1 viruses-13-02212-f001:**
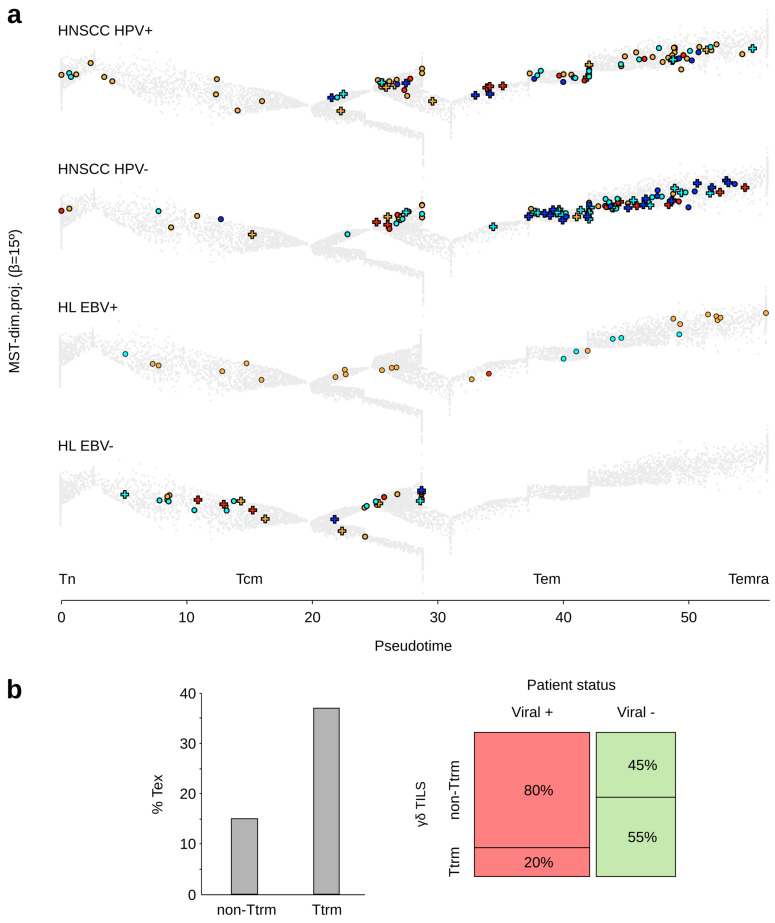
Differentiation and functional hallmarks of γδ T lymphocytes infiltrating HNSCC and HL tumors according to viral status of cancer patients. (**a**) Examples of γδ TILs extracted from representative HNSCC and HL tumors with viral status, overlaid on the public trajectory of γδ T lymphocytes reference (grey). Key: TCRVγ9 cells (light blue), Tex TCRVγ9 cells (dark blue), TCRVγnon9 cells (orange), Tex TCRVγnon9 cells (dark red), recirculating (non-Ttrm) cells (open circles), Ttrm (cross). Differentiation pseudotime scale: 0–5: Tn, 5–20: Tcm, 20–50: Tem; > 50: Temra. Together all the EBV-negative HL tumors totalized 53 mostly functional (43/53, 81%) γδ TILs, including 13 Ttrm with 39% of exhausted cells (5/13 Ttrm), while all the EBV-positive HL tumors totalized 60 mostly functional (56/60, 93%) γδ TILs, including 2 non-exhausted Ttrm γδ TILs. The HPV-negative HNSCC tumors totalized 698 mostly functional (489/698, 70%) γδ TILs, including 326 Ttrm with 39% of exhausted cells (127/326 Ttrm), while the HPV-positive HNSCC tumors totalized 357 mostly functional (310/357, 87%) γδ TILs, including 81 Ttrm with 31% of exhausted cells (25/81 Ttrm). (**b**) Right: rates of recirculating and Ttrm among γδ TILs (both subsets) per viral status of patients (pooled); left: rates of exhausted cells among the recirculating and Ttrm γδ TILs.

**Figure 2 viruses-13-02212-f002:**
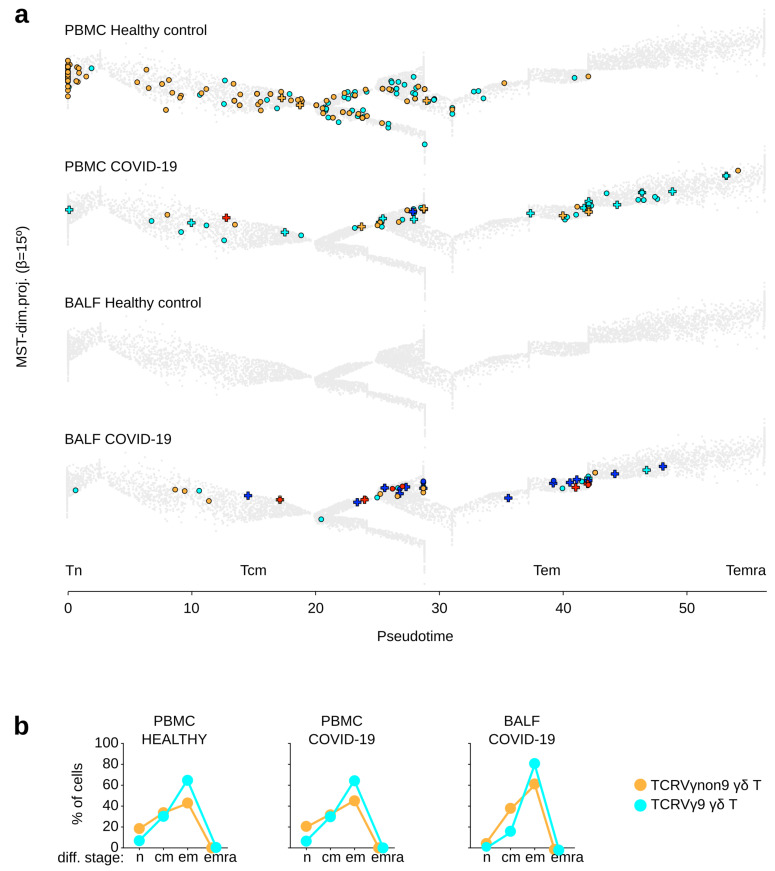
Differentiation and functional hallmarks of γδ T lymphocytes from control donor’s PBMC and BALF. (**a**) γδ T lymphocytes extracted from PBMC and BALF of healthy individuals and COVID-19 patients. No γδ T lymphocyte was found in any of the healthy control’s BALF (representative examples, cells are shown overlaid on the public trajectory of γδ T lymphocyte differentiation, same legend key as in Figure 1). (**b**) Rates of cells at each differentiation stage among the TCRVγ9 and TCRVγnon9 γδ T lymphocytes (group means are shown).

**Figure 3 viruses-13-02212-f003:**
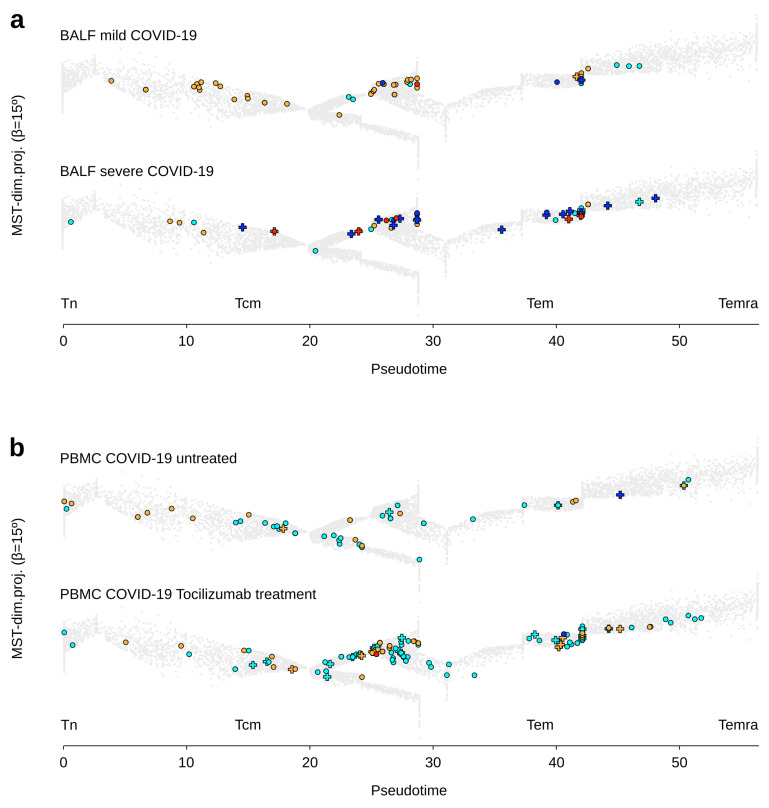
Differentiation and functional hallmarks of γδ T lymphocytes from adult COVID-19 patients. (**a**) BALF-derived γδ T lymphocytes from adult patients with mild or severe COVID-19 disease. (**b**) PBMC-derived γδ T lymphocytes from adult COVID-19 patients untreated or treated with Tocilizumab. (Representative examples overlaid on the public trajectory of γδ T lymphocyte differentiation, same legend key as in Figure 1).

**Figure 4 viruses-13-02212-f004:**
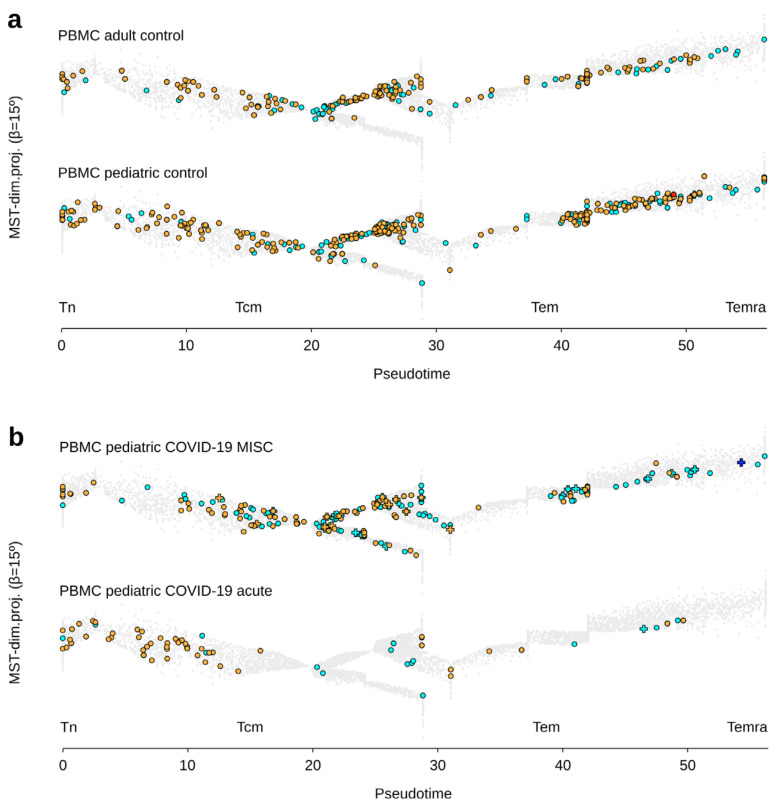
Differentiation and functional hallmarks of γδ T lymphocytes from pediatric COVID-19 patients. (**a**) PBMC-derived γδ T lymphocytes from adult and pediatric controls. (**b**) PBMC-derived γδ T lymphocytes from pediatric COVID-19 patients with either MIS-C or acute disease. (Representative examples overlaid on the public trajectory of γδ T lymphocyte differentiation, same legend key as in Figure 1).

## Data Availability

Further information and reasonable requests for resources and processed data, such as integrated (cell, UMI) matrices of extracted γδ T and their corresponding cell annotations, should be directed to J.-J.F. (jean-jacques.fournie@inserm.fr) or J.P.C. (juan-pablo.cerapio-arroyo@inserm.fr). Source scRNAseq datasets used in this study are available at GEO with the accession numbers: GSE139324, GSE145926, GSE155224, GSE155249, GSE166489, and GSE167029; and at the European Genome-Phenome Archive (EGA) with the accession number: EGAS00001004085. Single-Cell Multilayer Viewer (scMLV) is available on GitHub repository: https://github.com/MarionPerrier/scMLV (accessed on 31 March 2021). Single-Cell Signature Scorer and Single-Cell Virtual Cytometer are available at: https://sites.google.com/site/fredsoftwares/products (accessed on 12 October 2021). Seurat v4 is available from the Comprehensive R Archive Network (CRAN) and further details in the installation can be found on https://satijalab.org/seurat/ (accessed on 12 October 2021).

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
