# Peer review of "Single-Cell RNAseq Profiling of Human γδ T Lymphocytes in Virus-Related Cancers and COVID-19 Disease"

_viruses, 2021, doi:10.3390/v13112212_

Round 1

Reviewer 1 Report

The paper of Juan-Pablo Cerapio uses single-cell RNA-sequencing data to characterize gd T cells under different disease conditions. Therefore, gd T cells were digitally identified and characterized according to their ~ effector/memory differentiation and other characteristic profiles. gd T cells from HNSCC and HL were analyzed and compared according to their HPV and EBV status, respectively. In addition, samples from COVID-19 patients were analyzed.

However, as a reader, I have certain problems following the central idea of the paper. In the end, it did not really become clear to me what (except for the method) what is the connection between gd T cells in the context of tumor versus COVID-19. I would suggest separating both complexes.

Since the method of identifying gd T cells and their subpopulations on a single-cell base according to their transcriptome is a very abstract subject, it might improve the paper to elucidate the criteria according to which the populations were assigned to those groups in more detail. Also including data from the protein-level (flow cytometry), would improve and underline the overall message from this paper.

As the identification is based on data, is there a danger of erroneous grouping? In the PBMC samples, there are much more non-Vg9 T cells found than what would be expected from the normal PBMC distribution. Is there an explanation for this?

The statement of gd T-cell lymphodepletion in COVID-19 seems to be only based on the absolute cell number found in the respective samples, but it would be important to understand if the reduced numbers are specific for gd T cells relative to other immune cell populations and if this is statistically relevant.

Author Response

The paper of Juan-Pablo Cerapio uses single-cell RNA-sequencing data to characterize gd T cells under different disease conditions. Therefore, gd T cells were digitally identified and characterized according to their effector/memory differentiation and other characteristic profiles. gd T cells from HNSCC and HL were analyzed and compared according to their HPV and EBV status, respectively. In addition, samples from COVID-19 patients were analyzed.

1. However, as a reader, I have certain problems following the central idea of the paper. In the end, it did not really become clear to me what (except for the method) what is the connection between gd T cells in the context of tumor versus COVID-19. I would suggest separating both complexes.

The main idea of our paper was to report the first differentiation maps for gdT lyphocytes in human tumors of virally positive v.s. negative patients because our previous study (Cerapio et al. 2021) showing that the differentiation of these cells reflected those of T CD8 lypmphocytes from the same samples, raised the question of whether and how gd TILs diverged between virally-positive and -negative patients. In addition, since COVID-19 is a global health issue and a viral disease we thus applied the same analytic technology to depict the gd T cells in this disease.

2. Since the method of identifying gd T cells and their subpopulations on a single-cell base according to their transcriptome is a very abstract subject, it might improve the paper to elucidate the criteria according to which the populations were assigned to those groups in more detail. Also including data from the protein-level (flow cytometry), would improve and underline the overall message from this paper.

We agree with reviewer for this comments, since in the absence of our former, now published, manuscript (enclosed as PDF) which depicts in detail this issue, how populations were classified was hard to understand by readers. This is done as described in the parper’s Methods section “Cell classification”. Briefly, all classifications, including those of Ttrm v.s. recirculating and Tex v.s. functional by ‘at least one signature’ as follows. All single cells were scored for several reference gene signatures of Ttrm, recirculating, Tex, and functional (References and gene signatures are available in Cerapio et al 2021). For each signature, a cut-off was defined as the maximal score of the control γδ T lymphocytes extracted from healthy adults PBMC datasets. This threshold defined the cell’s binary (1: cell score > threshold, 0: otherwise). The Tex cut-offs were: 3.9 for ‘Chihara_IL27_Coinhib_module’; 0.52 for ‘Alfei_d20_tox’ ; 0.22 for ‘Khan_Tox_OverExpressed_genes’; 0.5 for ‘Tosolini_ NHL_IEGS33’; and 0.16 for ‘Balanca_QP_genes’. For each TIL, the five Tex binaries were summed, and any TIL with a non-zero sum of binaries was classified “Tex”, or “functional” otherwise. The Ttrm classification was applied likewise using six references and partially overlapping Ttrm signatures from the literature. The binarizing cutoffs established with the control γδ T lymphocytes were: 0.46 for ‘Kumar_13g_Ttrm’; 0.3 for ‘Kumar_3g_Ttrm’; 3.72 for ‘Wu_Tcellcluster4.1_trm’; 2.6 for ‘Wu_Tcellcluster8.3_trm’; 2.8 for ‘Wu_Tcellcluster8.3b_trm’; 4.2 for ‘Wu_Tcellcluster8.3c_trm’. These Ttrm binaries were summed, and any TIL with a non- zero sum of Ttrm binaries was classified Ttrm, or recirculating otherwise. Such a classification method has been formally validated for Tn, Tcm, Tem, Temra by our original CITE-seq analysis (scRNAseq + Immunophenotyping of the same single cells) of PBMC from healthy donors (Cerapio et al 2021; Figure S4A, S4B).

3. As the identification is based on data, is there a danger of erroneous grouping? In the PBMC samples, there are much more non-Vg9 T cells found than what would be expected from the normal PBMC distribution. Is there an explanation for this?

We have not a straightforward explanation for this. However, it is known that γd TCR expressed by γd T lymphocytes are either TCRVγ9/TCRVd2, TCRVγ9/TCRVdnon2, TCRVγnon9/TCRVd1, and TCRVγnon9/TCRVdnon1, with the first group generally prominent in blood and the others in tissues. We are very sorry that the currently available scRNAseq (either 3’ and 5’ chemistries) datasets never measure the TCRVd genes but only the constant segments of TCRγ chain (TRGC1-encoded TCRVγ9 and TRGC2-encoded TCRVγnon9). Thus, there is currently no means to better discriminate between TCRVγ9/TCRVd2, TCRVγ9/TCRVdnon2, TCRVγnon9/TCRVd1, and TCRVγnon9/TCRVdnon1, other than the mere binary classification into TCRVγ9 and TCRVγnon9 groups. The correspondence of the TCRVγ9 v.s. TCRVγnon9 grouping method with truly FACs-purified TCRVd1 and TCRVd2 γdT cell populations is largely consistent, as formerly assessed (Cerapio et al. 2021; Figures S1B, S1C). Although a risk of erroneous grouping exists but cannot be assessed directly, its definitive solution requires that the datasets from scRNAseq 5’ chemistry include the measure of all the TCRVd and TCRγ genes (which are not implemented currently).

4. The statement of gd T-cell lymphodepletion in COVID-19 seems to be only based on the absolute cell number found in the respective samples, but it would be important to understand if the reduced numbers are specific for gd T cells relative to other immune cell populations and if this is statistically relevant.

We thank Reviewer #1 for raising this interesting point. To examine whether this COVID-related lympho-reduction from PBMC is specific for gd T cells, we applied the corresponding digital purification pipeline for T CD8 cells from the same sample’s datasets. Their differentiation mapping is out of the scope of this study, so here we will only summarize the conclusion as follows: the PBMC samples from control donors and COVID-19 patients encompassed on average n= 479 and n= 561 T CD8 cells per sample, respectively, while the same groups had gd T cells counts of n= 330 and n= 100 cells. So the COVID-19-related lympho-reduction from PBMC is specific of the gd T but not T CD8 lineage (Chi2 P value= 5E-27). This interesting point is now added in lines 262-265 (text shown red) of the MS revised version.

Reviewer 2 Report

The study by Cerapio et al attempts to profile the heterogeneity of human TCRgd T cells in a range of disease models using established scRNAseq data sets, and importantly, integrate these data between new and old sets.  While this topic is interesting and relevant, I have concerns regarding the presentation and interpretability of the data:

  • The authors utilise one style of output/readout, yet the pseudotime plots are not always the most effective at conveying the message. In particular, quantification of subset frequency cannot be accurately shown on these graphs. Authors should show a plot with the individual data points and error bars
  • Could the authors explain the ‘pseudotime’ parameter and the allocation of Tcm/Tem/Temra across this axis?
  • While I appreciate the integration of data sets, could the authors explore additional methods to analyse the data to demonstrate the findings? Eg enrichment plots
  • In general, the manuscript is difficult to interpret and unclear in the main findings

Author Response

The study by Cerapio et al attempts to profile the heterogeneity of human TCRgd T cells in a range of disease models using established scRNAseq data sets, and importantly, integrate these data between new and old sets. While this topic is interesting and relevant, I have concerns regarding the presentation and interpretability of the data:

1. The authors utilise one style of output/readout, yet the pseudotime plots are not always the most effective at conveying the message. In particular, quantification of subset frequency cannot be accurately shown on these graphs. Authors should show a plot with the individual data points and error bars.

We apologize for not understanding what Reviewer #2 precisely refers to, here by “subset”. Is this subsetting binning about TCRVg9 versus TCRVgnon-9, or Tn, versus other differentiation stages (Tcm, Tem, Temra), or Ttrm versus Trecirculating, or Tex versus T functional ? The classic composite of differentiation stage frequencies per individual in each sample group are shown (means only) in Fig 2b for the TCRVg9 and TCRVgnon9 cells. For clarity of the figures here we deliberately favored showing only pseudotimed trajectory maps rather than the very classic barplots, largely described elsewhere.

2. Could the authors explain the ‘pseudotime’ parameter and the allocation of Tcm/Tem/Temra across this axis?

We agree with reviewer for this comment, since the method is hard to understand in the absence of our recently published manuscript (Cerapio et al, OncoImmunology 2021, enclosed as PDF), which depicts in detail this parameter and its method of computing. Pseudotime is technically the distance of each single cell to the trajectory’s first cell, named anchor. Since the whole trajectory represents the entire differentiation process, featured through the constituting dataset’s cells, each cell’s distance to the trajectory’s origin represents its pseudotime, which is a non-chronological representation of time. Simply speaking, the pseudotime’s origin and end could also be referred to as ‘ cell birth’ and ‘cell death’. The differentiation stages Tn,...Temra are defined in detail in (Cerapio et al, OncoImmunology 2021). These stages were defined on the basis of i) transcriptome correspondence with formerly depicted T cell stages signatures (refs 11-17 from the above cited Cerapio paper); ii) individual single cell scores for each of the gene signatures listed in Table S1, and summarized in the Figure 1.

3. While I appreciate the integration of data sets, could the authors explore additional methods to analyse the data to demonstrate the findings? Eg enrichment plots.

As replied above, in this report we deliberately opted to show only pseudotimed trajectory maps rather than more classic views such as enrichment plots. Indeed Signature enrichment plots give the same results. To illustrate this, we applied and showed (here below) the classical heatmap of TD-Enriched (TDE: trajectory-differentially enriched) genes shown clustered per ontology term, and gd T cell milestones (panel b), together with enrichment plots for selected gene signatures shown in function of pseudotime (panel c, lines shows ponderated smoothed means). Given the trajectory’s milestones identity (panel a), we think that the same results shown by pseudotimed representations (panel c) are more explicite to readers than when shown as ontology-clustered enrichment plots (panel b). (Figure in enclosed PDF)

4. In general, the manuscript is difficult to interpret and unclear in the main findings

The main idea of our paper was to report the first differentiation maps of human gdT lymphocytes in tumors of viral-positive v.s. negative patients because our previous study (Cerapio et al. OncoImmunology 2021; an MS showing that the differentiation of these cells reflected those of T CD8 lymphocytes infiltrating the same tumors) raised the question of whether and how gd TILs diverged between viral-positive and -negative patients. Since COVID-19 is a viral disease (and a global issue) which gd T cell compartment remains poorly characterized at the scRNAseq level, we thus applied the same analytic technology and pipeline to depict this latter.

Reviewer 3 Report

The findings appear to be interesting and technically well performed. Specific points that the authors need to address are as follows:

  1. How both the subsets of gd T lymphocytes can relocalize from PBMC to the infected lung tissues should be analyzed?
  2. The mechanism(s) by which anti-IL6R treatment can promote gd T lymphocyte differentiation should be analyzed.
  3. A limited in vivo study can be done to validate the reported findings.
  4. The authors should provide their own justification and relevance of the study. This will help the readers to understand the importance of the paper.
  5. Minor typographical errors were found throughout the manuscript and should be corrected.

Author Response

The findings appear to be interesting and technically well performed. Specific points that the authors need to address are as follows:

1. How both the subsets of gd T lymphocytes can relocalize from PBMC to the infected lung tissues should be analyzed?

We agree but this can not be done retrospectively for the published datasets. We are currently preparing such a study with a novel independent series of scRNAseq data from blood and tumors from the same lymphoma patients.

2. The mechanism(s) by which anti-IL6R treatment can promote gd T lymphocyte differentiation should be analyzed.

We agree but this cannot be done retrospectively from the published data. This is planned in the future as an independent study with newer datasets from multimodal single cell omics combining (scRNAseq from both 3’, and 5’ chemistries as well as CITE-seq and spatial transcriptomics)

3. A limited in vivo study can be done to validate the reported findings.

We agree with reviewer for this comments, since in the absence of the published manuscript (Cerapio et al, enclosed as PDF), which depicts in detail this issue in its results section “Ex vivo immunophenotype of gd T and T CD8 TILs are correlated”, illustrated in Figure 3.

4. The authors should provide their own justification and relevance of the study. This will help the readers to understand the importance of the paper.

The first-ever description of gd T lymphocytes by scRNAseq from both COVID-19 and virus-related cancer samples was initially proposed to the inviting editor (Dr. E. Champagne), so we have no plan nor means in a reasonable delay to re-investigate herein the same samples by additional different techniques.

5. Minor typographical errors were found throughout the manuscript and should be corrected.

We had the paper revised by a native english speaking editor, so these typos are now corrected.

Reviewer 4 Report

Authors investigated the single-cell RNAseq profiling of human γδ T lymphocytes in virus-associated cancers (head and neck squamous cell carcinoma with or without HPV, and Hodgkin's lymphoma with or without EBV) and COVID-19 infectious disease.   This type of approach (such as single cell RNAseq profiling) is quite new to know how the subpopulation of T cells were existing at the pathological sites. They found that  γδ-tumor infiltrating lymphocytes in virus-associated cancers were more prone to repeated circulation from tumor sites to avoid exhaustion. For COVID-19,  authors examined peripheral blood and BALF, comparison between adult and pediatric patients. Their findings were summarized in the "Abstract" like, "exhaustion. In COVID-19, both TCRVγ9 and TCRVγnon9 subsets of lymphocytes relocalize from PBMC to the infected lung tissue, where their advanced differentiation, tissue residency, and exhaustion reflect T cell activation. Although severe COVID-19 disease increases both recruitment and exhaustion of  γδ T lymphocytes in infected lung lesions but not blood, the anti-IL6R treatment with Tocilizumab promotes  T lymphocyte differentiation in COVID-19 patients. Pediatric patients with acute COVID-19 disease display similar γδ T cell lymphopenia from PBMC as the adult patients. However, blood  γδ T cells from pediatric COVID-19-related multi-system inflammatory syndrome in children (MISC) are not lymphodepleted, but differentiated as in healthy PBMC."   OK, indeed, it sometimes is still difficult to find significant differences among all the plotted figures. But, I can trust the findings written by authors and I could pursue the descriptions. Thus, I feel we can start this manuscript to consider the status of γδ T cells in virus-related pathological conditions such as cancers as well as lung infection like COVID-19.

Author Response

Authors investigated the single-cell RNAseq profiling of human γδ T lymphocytes in virus-associated cancers (head and neck squamous cell carcinoma with or without HPV, and Hodgkin's lymphoma with or without EBV) and COVID-19 infectious disease.   This type of approach (such as single cell RNAseq profiling) is quite new to know how the subpopulation of T cells were existing at the pathological sites. They found that  γδ-tumor infiltrating lymphocytes in virus-associated cancers were more prone to repeated circulation from tumor sites to avoid exhaustion. For COVID-19,  authors examined peripheral blood and BALF, comparison between adult and pediatric patients. Their findings were summarized in the "Abstract" like, "exhaustion. In COVID-19, both TCRVγ9 and TCRVγnon9 subsets of lymphocytes relocalize from PBMC to the infected lung tissue, where their advanced differentiation, tissue residency, and exhaustion reflect T cell activation. Although severe COVID-19 disease increases both recruitment and exhaustion of  γδ T lymphocytes in infected lung lesions but not blood, the anti-IL6R treatment with Tocilizumab promotes  T lymphocyte differentiation in COVID-19 patients. Pediatric patients with acute COVID-19 disease display similar γδ T cell lymphopenia from PBMC as the adult patients. However, blood  γδ T cells from pediatric COVID-19-related multi-system inflammatory syndrome in children (MISC) are not lymphodepleted, but differentiated as in healthy PBMC."   OK, indeed, it sometimes is still difficult to find significant differences among all the plotted figures. But, I can trust the findings written by authors and I could pursue the descriptions. Thus, I feel we can start this manuscript to consider the status of γδ T cells in virus-related pathological conditions such as cancers as well as lung infection like COVID-19.

We thank Reviewer # 4 for his positive comments on our work. The pseudotimed maps shown in these figures were selected as a representative example of the findings for each individual. The significant differences only emerged when comparing groups as a whole. We provide Reviewer #4 with the PDF of our recently published paper (Cerapio et al. OncoImmunology, 2021) which introduces pseudotimed trajectories.

Round 2

Reviewer 2 Report

The author's have addressed the original comments and therefore the manuscript adequate for publication.